# Self-Attention-Based Models for the Extraction of Molecular Interactions from Biological Texts

**DOI:** 10.3390/biom11111591

**Published:** 2021-10-27

**Authors:** Prashant Srivastava, Saptarshi Bej, Kristina Yordanova, Olaf Wolkenhauer

**Affiliations:** 1Institute of Computer Science, University of Rostock, 18059 Rostock, Germany; prashant.srivastava@uni-rostock.de (P.S.); saptarshi.bej@uni-rostock.de (S.B.); kristina.yordanova@uni-rostock.de (K.Y.); 2Leibniz-Institute for Food Systems Biology, Technical University of Munich, 85354 Freising, Germany

**Keywords:** text mining, self-attention models, biological literature mining, relationship extraction, natural language processing

## Abstract

For any molecule, network, or process of interest, keeping up with new publications on these is becoming increasingly difficult. For many cellular processes, the amount molecules and their interactions that need to be considered can be very large. Automated mining of publications can support large-scale molecular interaction maps and database curation. Text mining and Natural-Language-Processing (NLP)-based techniques are finding their applications in mining the biological literature, handling problems such as Named Entity Recognition (NER) and Relationship Extraction (RE). Both rule-based and Machine-Learning (ML)-based NLP approaches have been popular in this context, with multiple research and review articles examining the scope of such models in Biological Literature Mining (BLM). In this review article, we explore self-attention-based models, a special type of Neural-Network (NN)-based architecture that has recently revitalized the field of NLP, applied to biological texts. We cover self-attention models operating either at the sentence level or an abstract level, in the context of molecular interaction extraction, published from 2019 onwards. We conducted a comparative study of the models in terms of their architecture. Moreover, we also discuss some limitations in the field of BLM that identifies opportunities for the extraction of molecular interactions from biological text.

## 1. Why Text Mining?

Text mining techniques used for extracting information from text have been popularly used since 1992. Famous applications of text mining include IBM’s Watson program, which performed spectacularly when competing against humans on the nightly game show Jeopardy [1]. Such techniques have played a significant role over the years in extracting and organizing information from biological texts. For example, the popular STRING database [2] uses automated text mining of the scientific literature to integrate all known and predicted associations among proteins, including both physical interactions and functional associations [2].

Biological systems are complex in nature. Years of research have produced a large volume of publications on the key molecular players involved in numerous cellular processes, disease phenotypes, and diseases. For example, a PubMed search for the molecule “p53” produces more than a hundred thousand hits; a PubMed search for the disease “colorectal cancer” produces more than two-hundred thousand hits. For cell-level and tissue-level processes such as “apoptosis” and “metastasis”, there are more than four-hundred thousand hits.

In the five-year span of 2016–2020, the average number of PubMed article hits per year for “p53”, “colorectal cancer”, “apoptosis”, and “metastasis” were 4974, 13,548, 29,812, and 22,305, respectively. One obvious application for text mining is the search for information from the literature, as part of research projects. Since there are various databases and disease map projects that map out molecular interactions relevant to chosen diseases, the maintenance of such repositories requires substantial effort. A motivation for text mining is then also to assist the updating of data in repositories with new information from publications.

Modeling biological systems can have diverse motivations: investigating molecular interactions and their nature to understand regulatory mechanisms, investigating associations among molecules and diseases or broader disease phenotypes, investigating the consequences of genetic mutations and perturbations to cellular processes. Clearly, molecules such as genes, proteins, and drugs play a crucial role in such investigations. Rather than attempting to describe complex biological processes as a function of a handful of molecules, systems biologists increasingly appreciate the complexity of these systems, trying to visualize these processes as functions achieved through interactions among numerous molecular entities. These molecular entities (genes, proteins, or drugs) interact in harmony inside biological systems for each phenotype to be realized, be it a cellular process (e.g., cell signaling, metabolism, apoptosis), a disease phenotype (e.g., acute inflammation, metastasis), or even a disease (e.g., cancer, gaucher disease). However, a comprehensive systemic understanding requires extracting and integrating knowledge acquired from existing and new publications. In many cases, this results in large-scale models that require much manual effort from the modelers, who laboriously hand-pick knowledge from hundreds of publications [3].

Text mining and Natural-Language-Processing (NLP)-based techniques are finding their applications in reducing the efforts of biologists to mine the biological literature for tasks such as the creation of large-scale models and keeping databases updated. This recent field of research focused on automatic knowledge extraction and mining from biomedical literature is known as Biomedical Literature Mining (BLM). In this review article, we focus on recent developments in BLM for the extraction of molecular interactions from biological texts.

BLM consists of several types of tasks, combinations of which can be realized as complex workflows to achieve the goal of knowledge extraction. An elementary task, for instance, is Named Entity Recognition (NER), which aims to identify biomedical concepts from given text corpora. State-of-the-art models can perform this task with high accuracy. This upstream task is usually followed by Relationship Extraction (RE). Approaches for the RE task can be broadly categorized as rule-based approaches and Machine Learning (ML) approaches. Rule-based approaches depend on predefined rules based on inherent textual patterns in biomedical texts. The success of such approaches depends on the quality of the designed rules. In ML approaches, RE is usually posed as a classification problem. However, the design of the classification problem can vary with the motivation of the modeler. For example, there can be a binary classification problem that aims to model merely whether there exists an interacting pair of proteins in a document. More complicated multiclass classification problems investigate the nature of interactions among entities [4]. A general workflow for BLM is provided in Figure 1.

ML-based approaches that are used for RE have several broad categorizations such as feature-based approaches, kernel-based approaches, and neural-network-based approaches. Feature-based approaches involve the extraction of expert annotated lexical and syntactic features and use the same for modeling. Kernel-based approaches aim to map syntactic trees to higher-dimensional feature spaces by the proper choice of kernels. Neural-Network (NN)-based approaches can learn latent feature representations from labeled data. Neural network architectures such as Convolutional Neural Networks (CNNs), Recurrent Neural Networks (RNNs), Long Short-Term Memory (LSTM), and Gated Recurrent Units (GRUs) have been widely explored in this domain. Recent advances in the field of NLP are due to the introduction of a new class, known as self-attention-based models. These models account for long-range dependencies in text data and can learn contextual associations in text data better than previous neural-network-based models [5]. Using transformer architectures as the basis, several context-specific pretrained models such as BERT and BioBERT have been built, aimed at facilitating learning from biomedical texts [4,6,7].

Rule-based text mining approaches have been reviewed comprehensively by Zhao et al. [4]. Several deep-learning-based approaches were covered by Zhang et al. [6], covering publications until 2019. Self-attention-based models entered the stage around 2017; pretrained networks for the extraction of molecular interactions at an abstract or sentence level, from biomedical texts, such as BERT and BioBERT, were published after 2018. In this review, we therefore focus on self-attention-based models, both novel architectures and pretrained networks, published from 2019 onwards. For mining recent works on self-attention-based models, we searched for published articles and preprints that were publicly available. We searched for well-known databases for publications such as https://arxiv.org/ arXiv, https://www.biorxiv.org bioRxiv, https://pubmed.ncbi.nlm.nih.gov/ PubMed, https://www.semanticscholar.org/ Semantic Scholar, etc. In addition to these, we also searched for publication databases on recent issues of several related journals and conferences. We only surveyed research articles that involved a self-attention mechanism for biological literature mining. We first give a brief description of the philosophy behind self-attention. Next, we discuss the architectural aspects and compare the performances of some recent models proposed in the context of BLM. Finally, we discuss the pros and cons of using such models in the context of BLM before concluding our article.

Note that, given that there are many abbreviations for the terminologies relevant to this article, we provide some important abbreviations in Table 1, for the convenience of the readers.

## 2. Evolution of Deep Learning Models for NLP

Sequence-to-sequence models typically receive a sequence as the input and generate a sequence as the output. Input and output sequences can be numerical, time-dependent data, or string data. The Recurrent Neural Network (RNN) is a deep-learning-based model designed for learning from sequence data. At every learning step, RNNs take elements of a sequence as the input, generate an output for that time step, and update a hidden state that can be associated with the “memory” of the network. For text-based data, RNNs once used to be the state-of-the-art models. However, RNNs proved to be less effective to learn from longer sequences, that is to create associations among elements of long sequences. This means that if there is a long sequence of text (a long sentence) and there is an association between two words, one located at the beginning of the sentence and the other towards the end, RNNs are unlikely to capture that information. LSTMs and GRUs were designed to mitigate this “memory” problem. The extremely popular LSTM model, for example, is designed to retain or forget information that is stored in the hidden state sequentially. Transformers, in contrast to the previous models, receive the whole sequence as the input rather than taking elements of a sequence sequentially as inputs. To allow the model to recognize the sequential nature of the data, it employs the concept of positional encoding. The attention mechanism is then used to learn associations among elements of the sequence, which in turn are used to make decisions. Taking the entire sequence as the input helps this model learn relatively long-range associations among elements of a long sequence, which makes it apt for text data and thus applicable to NLP. Since the introduction of the attention model by Bahdanau et al. for machine translation in 2015, it has found applications in a wide range of NN-based architectures [8], while it received more recognition after the introduction of transformer models in 2017 [5]. However, apart from NLP, the attention mechanism has been applied in computer vision, time series analysis, and reinforcement learning [9,10,11]. In the NLP domain, attention models have helped improve machine translation, question-answering problems, text classification, representation learning, and sentiment analysis [12,13,14,15,16]. In what follows, we discuss some interesting aspects of the self-attention-based models. We briefly visualize the evolution of sequence-to-sequence models in Figure 2.

### 2.1. Self-Attention and Its Advantages

A typical sequence-to-sequence model consists of an encoder–decoder architecture [17]. The traditional encoder–decoder framework used in RNN, LSTM, or GRU has two main limitations, as mentioned in Chaudhari et al. [18]:The encoder compresses all input information into a vector of fixed length, which is passed to the decoder, causing significant information loss [18];Such models are unable to model the alignment between input and output vectors. “Intuitively, in sequence-to-sequence tasks, each output token is expected to be more influenced by some specific parts of the input sequence. However, decoder lacks any mechanism to selectively focus on relevant input tokens while generating each output token” [18].

The attention model tackles this issue by enabling the decoder to access the whole encoded sequence. The attention mechanism assigns attention weights over the input sequence, which captures the importance of each token in a sequence and prioritizes them for generating output tokens at each step.

The concept of self-attention came into prominence after the introduction of the transformer model. “Intra-attention, also known as self-attention, is an attention mechanism relating different positions of a single sequence to compute a representation of the sequence” [5]. Vaswani et al. demonstrated that the transformer architecture has a shorter training time and higher accuracy for machine translation without any recurrent component [5]. Transformers have become a state-of-the-art approach for NLP tasks, and they have been adopted for a variety of NLP problems such as the Generative Pretraining Transformer (GPT, GPT-2) for language modeling, the universal transformer for question answering, and Bidirectional Encoder Representations from Transformer (BERT) for language representation [15,19,20]. The transformer model has two key aspects:**Positional encoding:** Given an input sentence in a transformer model, the model first creates a vectorized representation of the sentence *S*, such that each word in the sentence is represented by a vector of a user-defined dimension. The vectorized version of the sentence *S* is then integrated with positional encoding. Recall that, unlike sequence-to-sequence models such as RNNs and LSTMs, which would feed the sequence elements (words in a sentence) as the input sequentially, self-attention-based models feed the entire sequence (sentence) as the input at a time. This requires a mechanism that can account for the sequential structure of the input sequence/sentence. This is achieved through positional encoding. The formal expression for positional encoding is given by a pair of equations:
(1)$$P_{\left(pos,2i\right)}=\sin\left(\frac{pos}{10000^{\frac{2i}{d}}}\right)$$
(2)$$P_{\left(pos,2i+1\right)}=\cos\left(\frac{pos}{10000^{\frac{2i}{d}}}\right)$$In Equations (Equation 1) and (Equation 2), the expression *pos* is used to denote the position of a word in a sentence and *d* denotes the dimension of user-defined dimensions for the word embeddings, that is each word is essentially perceived by the model as a *d*-dimensional vector. The index *i* runs over the dimensions of these word embeddings and take values in the range [1, d]. Note that Equations (Equation 1) and (Equation 2) propose two different functions over the vector, depending on whether one is calculating an odd index or even index of the word-embedding vector. The dependence of the positional encoding functions on $\frac{2i}{d}$, given that these functions are periodic functions by design, ensures that several frequencies are captured over several dimensions of the word-embedding vectors. “The wavelengths form a geometric progression from 2π to 10,000 × 2π” [5]. Intuitively, proximal words in a sentence are likely to have a similar *P* value in a lower frequency, but can still be differentiated in the higher frequencies. For far apart words in a sentence, the case is just the opposite. Equations (Equation 1) and (Equation 2) also ensure the robustness and uniformity of the positional encoding function *P*, over all sentences, independent of their length [5];**The self-attention mechanism:** Once the positional encoding is integrated with the word embedding of an input sentence *S*, the resultant vector *W* is fed into the mechanism of self-attention. There is a popular analogy used by many data scientists to explain the concepts of Query (*Q*), Key (*K*), and Value (*V*), which are central to the idea of self-attention. When we search for a particular video on YouTube, we submit a query to the search engine, which then maps our query to a set of keys (video title and descriptions), associated with existing videos in the database. The algorithm then presents to us the best possible values as the search result we see. For a self-attention mechanism [5],
(3)K=Q=V=WA dot product between *Q* and *K* in the form $Q\cdot K^{T}$ can measure the attention between pairwise words in a sentence, to generate attention weights. The attention weights are used to generate a weighted mean over the word vectors in *V*, to obtain relevant information from the input as per the given task. As these vectors are learned through the training procedure of the model, the framework can help the model retrieve relevant information from an input, for a given task. The equation governing the process is given by [5]:
(4)$$A\left(K,Q,V\right)=\mathrm{softmax}\left(\frac{K\cdot Q^{T}}{\sqrt{d}}\right)V$$In practice, however, a multiheaded attention mechanism is used. The idea of multiple heads is again often compared to the use of different filters in CNNs, where each filter learns latent features from the input. Similarly, in multiheaded attention, different heads learn different latent features from the input. The information from all heads is later integrated by a concatenation operation. To account for multiple heads, Equation (Equation 3) is violated of course, and the dimension of the positionally encoded word vector *W* is distributed over the multiple heads. Equation (Equation 4) is also adjusted accordingly by replacing the denominator of $\frac{K\cdot Q^{T}}{\sqrt{d}}$ by $d_{k}$, where $d_{k}$ is the dimension of the keys considering multiple heads. Several other concepts such as layer normalization and masking are also used in transformer models, which we will not discuss in detail here. A representation of the transformer architecture and attention map over a sentence is provided in Figure 3 [5].

### 2.2. Pretrained Models

Pretraining models has been in existence for a long time. The idea behind pretrained language models is to create a black box that can understand a language and can be used for specific tasks in that language. These language models are usually pretrained on very large datasets to generate embeddings, which are used in various NLP models. These learned word embeddings are generalized and do not represent any task-specific information. Hence, to utilize them properly, they are fine-tuned on task-specific datasets. Using these pretrained language representations can help decrease the model size and achieve state-of-the-art performance.

BERT was introduced in 2019 by Devlin et al., which is a bidirectional pretrained transformer network, trained on unlabeled texts. BERT aims to generate a language representation by utilizing the encoder network of the transformer model. BERT can be used in a variety of NLP tasks such as question-answering, text classification, language inference, sentiment analysis, etc. The pretrained BERT model can be fine-tuned with one additional output layer to create NLP models without requiring task-specific architecture engineering [15].

The BERT’s authors presented two BERT models, BERT_BASE_ and BERT_LARGE_. BERT_BASE_ consists of 12 transformer blocks, 12 self-attention heads, hidden units of size 768, and a total of 110M trainable parameters, whereas BERT_LARGE_ has 24 transformer blocks, 24 self-attention heads, with a hidden unit size of 1024 and a total of 340M parameters. BERT can take as the input both a single sentence and a pair of sentences as one token sequence, allowing it to handle a variety of NLP tasks. The first token of every sequence is a classification token ([CLS]). To separate sentence pairs, a token ([SEP]) is used. Moreover, a learned embedding is added to every token, indicating that it belongs to a sentence. The input representation is obtained by adding token embeddings, sentence embeddings, and positional embeddings.

Devlin et al. used two pretraining strategies for BERT: the first is the Masked Language Model (MLM), and the second is Next Sentence Prediction (NSP). The masked language model randomly chooses 15% of the input tokens and masks them by replacing the chosen tokens with the [MASK] token. These masked tokens are then predicted by BERT based on the context of other nonmasked tokens. The MLM task enables bidirectional transformer pretraining, which allows the model to learn the context of a word based on both its left and right surrounding words [15]. In the next sentence prediction task, the model receives pairs of sentences as an input and predicts whether the first sentence is followed by the second sentence. When choosing sentences *A* and *B* for the NSP pretraining task, 50% of the pretraining examples are chosen such that *A* is followed by *B* and labeled as IsNext. The other 50% of pretraining examples are chosen such that *A* is not followed by *B* and labeled as NotNext. For the pretraining corpus, the authors used BooksCorpus having 0.8 B words and text passages of English Wikipedia having 2.5 B words [21]. WordPiece embedding was used to create a vocabulary of 30,000 words [22]. BERT obtained state-of-the-art performance on eleven NLP tasks including the General Language Understanding Evaluation (GLUE) benchmark, the Stanford Question Answering Dataset (SQUAD), and the Situations With Adversarial Generations (SWAG) dataset [23,24,25].

Since BERT’s release, several BERT-based models have been released for domain-specific tasks, for example ALBERT, BERTweet, CamenBERT, RoBERTa, SciBERT, and BioBERT. BioBERT, presented by Lee and Yoon et al., is a pretrained language model for biomedical text mining [26,27,28,29,30,31]. During pretraining, BioBERT was initialized with weights from BERT and then trained on biomedical domain corpora. The biomedical corpora consisted of PubMed abstracts having 4.5B words and PMC full articles having 13.5B words. To ensure BERT’s compatibility with BioBERT, the original vocabulary of BERT was used. WordPiece tokenization was applied for words that were not present in BERT’s vocabulary (for example, immunoglobulin was tokenized as I ##mm ##uno ##g ##lo ##bul ##in) [22]. BioBERT outperformed BERT and other state-of-the-art models in three biomedical NLP tasks: NER, RE, and QA. BioBERT achieved state-of-the-art performance, requiring only minimal architectural modification. Since its introduction, BioBERT has been used in various NLP tasks [30].

## 3. Applications of Self-Attention-Based Models in BLM

### 3.1. Commonly Used Datasets

Interactions among genes, proteins, chemicals, and drugs is a well-explored field. These types of studies have been one of the cornerstones of systems biology, as they help visualize complex biological processes at a higher level of complexity. As a result, there are quite a few well-maintained and organized databases in these directions. As we observed in our review, the most popular ones used for self-attention-based models are the BioGRID, IntAct, DrugBank, and ChemProt datasets. In addition, many other PPI-based databases such as STRING, MINT, BIND, TRRUST, and AIR are publicly available [32,33,34,35,36,37]. These databases contain annotations for numerous proteins and interactions. Interestingly, however, these datasets are all annotated differently. For example, for the BioGRID database, there are fifteen types of annotated interactions: direct interaction, synthetic lethality, physical association, association, colocalization, dosage lethality, dosage rescue, phenotypic enhancement, phenotypic suppression, synthetic growth defect, synthetic rescue, dosage growth defect, negative genetic interaction, synthetic haploinsufficiency, and positive genetic interaction [32]. In contrast, for datasets such as TRRUST or AIR, there are only three types of mentioned interactions: activation (positive), repression (negative), and unknown (undefined). Some works also prefer to curate customized datasets for their studies [3,38]. Elangovan et al. considered the IntAct database as the basis of their training data creation. Their annotation was based on chemical characterizations of the interactions [33,39]. They designed their study as a classification problem on eight classes: acetylation, methylation, demethylation, phosphorylation, dephosphorylation, ubiquitination deubiquitination, and negative. Su et al. and Giles et al., on the other hand, used two and five types of annotations respectively for PPIs. Moreover, as Giles et al. explored in their study, even for human-annotated data, ambiguities persist [40].

### 3.2. Architectural Comparison of Some Recent Attention-Based Models

A summary of all discussed models is provided in Table 2. We now discuss the architectural aspects of the models in detail.

Elangovan et al. (2020) [39]: The motivation of the work by Elangovan et al. lied in the fact that in popular PPI databases such as IntAct, despite containing a large amount of information on PPIs, only 4% of these interactions are functionally annotated. The functional annotations of two interacting proteins can however be found in relevant publications. Given relevant text data (e.g., abstracts of publications), Elangovan et al. focused on extracting functional annotations of interacting proteins [39].

For this particular work, the authors selected PPIs from the IntAct dataset having seven types of functional annotations, namely: phosphorylation, dephosphorylation, methylation, demethylation, ubiquitination, deubiquitination, and acetylation. The task addressed in the article was, therefore, to determine the type of PPIs, rather than solely to determine whether two proteins interact. PPIs for which the type of interaction is explicitly mentioned in the abstract of a relevant article were termed as *typed interactions* [39].

Assuming that the type of the interaction of a PPI can appear anywhere in the abstract, possibly across multiple sentences, the authors used an abstract-level annotation of the PPIs. Due to this coarse-grained annotation method, where the data are labeled as per the co-existence of the PPI and the interaction type word in an abstract and not by precise causation between the two entities, the model was described by the authors as a “weakly supervised” one [39].

The authors were also careful to state their assumption that the annotated PPI be described in the abstract of the article, although in practice, this information may prevail in any part of the text. It was further assumed that if, for an annotated PPI in the IntAct database, the type of interaction does not appear in the abstract, then it is annotated somewhere in the full text. Such data instances motivated the authors to define negative samples in the training data. Given a protein pair $\left(p_{1},p_{2}\right)$, if there is no associated interaction word in the abstract against the IntAct annotation(s) of the pair, then the protein pair and the IntAct annotation form a negative sample. Note that this implies that a negative sample does not necessarily mean that the protein pair does not interact with each other, but merely that the abstract of the relevant article does not mention this interaction. This rather strong assumption also makes the data noisy, as mentioned by the authors. This, on the other hand, implies that *untyped interactions*, or interactions whose type is not known, would also be a subset of the negative samples [39].

The model used by the authors for this paper was a fine-tuned version of the BioBERT model. The fine-tuning process enabled BioBERT to adapt to the typed PPI classification task. The authors referred to this model as PPI-BioBERT in the article. To further improve the probability estimate of each prediction, the authors used an ensemble of 10 PPI-BioBERT models for decision-making [39].

Giles et al. (2020) [40]: While conventional string matching is used to search for co-occurrences of entities (gene or protein names) in a sentence, this results in the inclusion of large amounts of noise in the results. For instance, as the authors of this particular research work pointed out, in the case of the PPI detection problem, about 75% of the sentences containing co-occurring names of possibly interacting proteins do not describe any causal relationship among them. With this motivation, the authors investigated the possibility of using fine-tuned BioBERT to analyze these co-occurrences and thereby to accurately determine the functional association among the co-occurring proteins in a given sentence [40].

An interesting experiment conducted by the authors during the data preparation was the investigation of interannotator agreement. Three independent expert curators curated PPIs from 925 sentences identified by NER tagging within papers drawn from MEDLINE. Surprisingly, concordance among all three curators was observed in only 48.8% of the cases, which demonstrated the complexity of the problem [40].

Moreover, the authors experimented with the need for a semisupervised preprocessing step for training data curation. This experiment was necessary due to an inherent class imbalance between positive protein interactions and the coincidental mention of proteins. The authors repeated the data curation step after filtering the sentences such that only those that contained two genes identified to have a strong likelihood of interacting, signified by a high combined StringDB score, were retained. Even with high reliability scores from StringDB, no improvement in the rate of identification of positive interactions was found. However, for some other cases, such as the drug–drug interaction problem, this step proved to be more effective. The authors concluded that this type of preprocessing approach can assist in cases of balanced training data curation in specific problems.

As far as predictive models are concerned, the authors compared some rule-based approaches with a fine-tuned version of BioBERT [40].

Su et al. (2020 and 2021) [41,42]: We now discuss two research papers that are related to each other and share two common authors. The first paper investigated the scope of the BERT and BioBERT model in general BLM problems. The second paper improved on the result of the first one by improving the performance of the pretrained BERT model by using a pretraining step involving contrastive learning. Both papers used very similar study designs. The effectiveness of the models was demonstrated by applying them to three types of RE tasks from the biomedical domain: chemical–protein (ChemProt, using the BioGRID database), drug–drug (DDIs, using the DrugBank database), and protein–protein interactions (PPIs, using the IntAct database). The PPI classification task is considered a binary classification, indicating that the authors refrained from a more function-oriented classification, as explored by Elangovan et al., whereas the ChemProt and BioGRID classification tasks are multilabel classification tasks with five and four annotated interaction types in the respective databases [41,42].

In the first paper, Su et al. (2020) proposed some new fine-tuning mechanisms for the BERT model. They pointed out that the RE problems are posed as classification problems and pretrained models such as BERT rely on a specific [CLS] token from the last layer to make decisions. “The [CLS] token is used to predict the next sentence (NSP task) during the pretraining, which usually involves two or more sentences, but the inputs of our relation extraction tasks only contain one sentence. This indicates that the [CLS] output might ignore important information about the entities and their interaction because it is not trained to capture this kind of information [41]”. As a solution to this, the authors proposed to add a new module that could summarize all outputs from the last layer and concatenate that information with the [CLS] output as an extra fine-tuning step. The authors experimented with the choice of the new module used to summarize information using LSTM and additive attention [41].

In the second paper, Su et al. (2021) proposed a contrastive-learning-based approach to improve the performance of the pretrained models. The term contrastive learning is used for a family of methods to construct a discriminative model comparing pairs of inputs. The training process for such models is designed such that similar input instances have “positive” labels, whereas dissimilar input instances are labeled as “negative” instances. The goal is to learn a text representation by maximizing the agreement between inputs from positive pairs via a contrastive loss in the latent space, and the learned representation can then be used for relation extraction. The authors pointed out the lack of exploitation of the potential of such contrastive models for text data in general and RE problems from biomedical natural language processing specifically. The reason behind this, as explained by the authors, is that it is more challenging to design a general and efficient data augmentation method to construct positive and negative pairs necessary to train such models [42].

Moreover, in Su et al. (2021), the authors proposed a new metric, “prediction shift”, to measure the sensitivity degree to which the small changes of the inputs will make the model change its prediction, thereby arguing that the proposed model is more robust compared to simply using BERT for the classification of interaction words [42].

To generate a positive pair of samples compatible with the training design of the contrastive model, the authors resorted to simplistic data augmentation techniques. The goal was to slightly alter the original sentence using methods such as synonym replacement, the random swap of words, or the random deletion of words. Given a sentence *s*, two entity mentions (chemical or gene names) $e_{1}$ and $e_{2}$ in *s* and a relation type *r* also mentioned in *s*, the authors hypothesized that the Shortest Dependency Path (SDP) between the two entity mentions ($e_{1}$ and $e_{2}$) in the sentence *s* captures the required information to assert the relationship of the two entities. Keeping the SDP fixed, the authors therefore altered the rest of the word tokens in the text to generate augmented data, to ultimately generate positive samples. The hypothesis related to SDP is not novel in itself and has been explained in related research articles: “If entities $e_{1}$ and $e_{2}$ are arguments of the same predicate, then the shortest path between them will pass through the predicate, which may be connected directly to the two entities, or indirectly through prepositions.” Given a training batch of *N* sentences, the authors created an alternative “view” of each sentence (making a pool of 2N sentences), and then for every sentence *s*, they considered $<s,s^{\prime }>$ as a positive pair. The other 2N−1 sentences were considered to be a negative sample, each compared to the sentence *s* [42].

The general architecture of the model is fairly similar to the general structure of Siamese neural networks. Training samples (sentences) are fed into the neural network in pairs (labeled positive or negative), and each input sentence in the pair goes through two independent channels of an identical architecture. The final output is then generated by combining the outputs from these two independent channels, which are used to calculate the loss, which is optimized to be less for similar sentences (positive pairs). Each independent channel has a neural network encoder used to create encoding for the input sentences corresponding to the channel and a projection head (a multilayered perceptron) to transform the encoding to a desired dimension, which is known to improve the representation quality during training [42].

Wang et al. (2020) [43]: RE among proteins is affected by mutations, implying that interactions among proteins may vary from one study to another depending on these mutations, as well as the context of the study. To this end, the Biocreative VI challenge consists of two subtasks:Identifying documents describing mutations affecting PPI;Extracting relevant PPI through RE.

The first task, also referred to as document triage by the authors, clearly improves the practicality of using NLU-based models for RE in the context of PPI. The second task can extract interacting protein pairs from documents containing a triage. The term “triage” refers to a tuple of a source protein, a target protein, and their relevant interactions. Although RE is the main task addressed in this research article, the authors argued that the introduction of auxiliary tasks, such as document triage classification (whether a document describes genetic mutations affecting protein–protein interactions) and the gene recognition task (NER), significantly improves the RE task [43].

The experiments for triage and RE tasks were performed on the BioCreative VI Track 4 corpus, containing 4082 articles in the training set, of which 1729 were relevant to PPIs involving mutations. Standard preprocessing approaches such as replacing mentions of gene names by predefined strings were employed [43].

The architecture of the model is compatible with the multitask (main and auxiliary tasks) learning strategy as proposed by the authors. For creating meaningful vector representation of the input text, the authors used the BERT and BioBERT models. The BERT layer was shared as an embedding layer for all downstream layers. For the main RC task and auxiliary document triage task, a downstream text CNN model was added to the model. Independent BiLSTM layers were used as a downstream layer for the gene recognition auxiliary task. The authors argued that the introduction of the auxiliary learning tasks improved the classification performance of the main RE task [43].

Zhou et al. (2019) [44]: In this research work, the authors proposed the Knowledge-aware Attention Network (KAN) for PPI extraction. The motivation of this work, published in 2019, was the fact that pre-existing methods needed extensive feature engineering and could not make full use of the prior knowledge available in the form of knowledge bases [44].

Experiments with the model were conducted on the BioCreative VI Track 4 PPI extraction task corpus. PPI relation triplets were extracted from two knowledge bases, IntAct and BioGRID, both of which contain 45 relation types. A total of 1,518,592 triples and 84,819 protein entities were obtained for knowledge representation training, i.e., they were fed as prior knowledge to the model during training. As other approaches, the KAN model has elaborate preprocessing protocols. Some assumptions adopted during the preprocessing seem to be rather strict. For example, the authors wrote: “To reduce the number of inappropriate instances, the sentence distance between a protein pair should be less than three.” In addition to this, other general protocols such as replacing gene/protein names and context-specific words (interactions) by predefined strings were also employed [44].

As far as the model architecture is concerned, KAN is innovative. A schematic representation of the model is shown in Figure 4. KAN has two architectural components that are identical in structure, one for processing information relevant to a source protein and the other for processing information relevant to the corresponding target protein, given a source–target protein pair in a sentence. The information on the positions of the source and target proteins is encoded along with the sentence while the input is fed into the model. This is performed by modifying the general idea of positional embedding that is employed in the self-attention-based model in general. In this case, position encoding is encoded with respect to the positions of the source and target proteins in a given sentence. Respective positional encodings for the source and target proteins are fed into the respective architectural components along with the encoded sentence. Next, these inputs are passed through a diagonal-disabled multiheaded attention layer in each architectural component. Generally, self-attention-based processes are represented as some mathematical operations among a Query (*Q*), Key (*K*), and Value (*V*) vector. The same vector (vectorized form of the word sequence in a sentence added to the positional encoding) is considered for *Q*, *K*, and *V*. The multiplication of *K* and *V* produces a square attention matrix, which is then multiplied by *Q*. In the KAN model, however, the authors used a different form of *Q*. As the model aims to exploit the entity relation triplets recorded in triplets as prior knowledge, TransE (a typical knowledge representation approach, which represents the relation between two entities as a translation in a representation space) was used to create vector representations of these triplets. The vector representations of the source and target ($e_{1}$ and $e_{2}$) proteins were used as a part of *Q* in the respective architectural components. After passing the outputs of the attention layers through a feed-forward network and a multidimensional attention layer in each architectural component, the outputs from two architectural components were concatenated to obtain the final feature representation. At this stage, the vector representation of the relation between the source and the target proteins ($e_{r}$) was also concatenated with the vector representation of the proteins to take advantage of prior knowledge. The results from this layer were then passed through a softmax activation to obtain the final outputs for the classification task. The authors also experimented with several variations of the KAN model [44].

### 3.3. Performance Comparison among the Discussed Models

The models we discussed in the previous section were designed to perform diverse tasks varying from document triage finding to RE problems (PPI, DDI, ChemProt, etc.) to detection of “typed” PPIs. Moreover, they operate on different datasets and have different preprocessing approaches involved. In addition to these factors, they are also often evaluated on different performance measures. It is therefore difficult, if not impossible, to come up with a fair way of comparing their performances. However, one can still observe patterns in the results, which can be of significance.

The results of Wang et al. and Zhou et al. are comparable. Both of them addressed the same dataset, that is the BioCreative VI dataset. An evaluation criterion called “exact match evaluation” was also similar in both cases. It is defined as: “A predicted relation only counts when the GeneIDs are the same as human-annotated GeneIDs.” In this regard, Wang et al.’s model with an F1 score of 43.14 clearly outperformed the KAN model by Zhou et al., with an F1 score of 38.23, which confirms that learning auxiliary tasks along with the principal task could play a role in improving model performances. It is also noteworthy that the preprocessing protocols of Wang et al. were comparatively simpler. Although these two papers dealt with the extraction of interacting protein pairs from documents, they did not emphasize specifying the type of interaction. Knowing the type of interaction can be extremely useful while creating a large-scale disease map such as the Atlas of Inflammation Resolution [3] or the Parkinson’s Disease Map [45].

Elangovan et al., on the other hand, addressed the type of interaction among protein pairs. Their “typed” classification problem deals with seven different kinds of interactions: phosphorylation, dephosphorylation, methylation, demethylation, ubiquitination, deubiquitination, and acetylation. Thus, their model is a multiclass classification model PPI-BioBERT, which produced an F1 score of 35.4%. Interestingly, an ensemble of 10 PPI-BioBERT models improved the F1 score to 54%, which shows that there is a scope for ensemble-based models in RE-based problems [39].

Su et al. (2020 and 2021) explored the same three BLM-based problems, PPI, DDI, and ChemProt. The first article written in 2020 compared several variations of the BioBERT model (such as using LSTM and attention layers and utilizing the classification token [CLS] in their model). Using their models, they achieved F1 scores of 82.8, 80.7, and 76.8 for the PPI, DDI, and ChemProt tasks, respectively. However, from the results, it is difficult to conclude which kind of architecture (using LSTM or attention layers) in particular is beneficial. Perhaps that is why, in the follow-up article in 2021, the authors used none of the ideas of attention or LSTM layers. Rather, they focused on contrastive learning. The best results were produced by a model that used contrastive learning in addition to adding information from external knowledge bases. In this case, they achieved F1 scores of 82.7, 82.4, and 76.9 for the PPI, DDI, and ChemProt tasks, respectively, which was not a significant improvement on the previous work [41,42].

## 4. Discussion

The classification approach popularly used for RE hinders the transfer of knowledge across databases and corresponding datasets. This is due to the different annotations used in different databases. Knowledge transfer and integration across databases, in accordance with the classification approach, therefore requires the amalgamation of corresponding datasets with different annotations. This can create a multilabel classification problem with multiple classes, making a model difficult to train. We observed that the F1 score is naturally less in classification problems with multiple classes. For example, Su et al. had a binary classification task for PPI and thus had a superior F1 score compared to Elangovan et al., who considered multiple classes in their classification problem [39,41,42]. Furthermore, F1 scores are low for models that use exact match evaluation for measuring their performance. Intrinsic imbalance persistent in such datasets along with ambiguous manual annotations across datasets make it even harder to effectively train classification models. This reduces the practical usability of the classification models as a modeler cannot customize these models as per his/her modeling needs, and he/she have to rely on pre-annotated databases to train his/her models.

Moreover, modeling directed interactions requires knowledge on the source entity, the target entity, and the relationship between them. Usually, this used to be designed as a relationship-triplet-finding problem. Most classification models for RE do not consider the sense of directionality that is associated with the related entities. Only a few models, such as the KAN, consider taking the source and target entities as a part of the input and try to predict the corresponding interaction. However, still, the KAN is not trained in such a way that it can differentiate between a source and a target entity in the case of a directed interaction.

Even if such a classification model exists that can differentiate between a source and target entity of a directed interaction, while practically using such a model to extract relationships from new data, a modeler has to know from a relevant text which entity is the source and which entity is the target. Without knowing this information, directed interactions cannot be modeled. While NER-based models can identify the entity names from new literature, the issue of annotating source and target entities was not addressed in the discussed approaches, in general. This again hinders the practical applicability of such models in knowledge extraction.

What makes the practical use of many of these models difficult are the diverse preprocessing protocols and strict assumptions adopted by the models. Almost every model that we discussed replaces protein or chemical names by specific strings (e.g., Su et al., Wang et al., Elangoven et al., Zhou et al.) [39,41,42,43,44]. The model by Zhou et al., for example, adopted elaborate protocols while curating training data such as [44]:Reducing the number of inappropriate instances; the sentence distance between a protein pair was assumed to be less than three;Selecting the words among a protein pair and three expansion words on both sides as the context word sequence with respect to the protein pair;Removing protein names from the input string;Replacing numeric entries by predefined strings;

For attention-based models, even though there have been some attempts at making the models explainable by observing the attention matrices, such attempts are rare in the case of BLM. For example, Su et al. (2020) and Zhou et al. made some limited efforts to explain the behavior of their models [41,44].

In the field of computer vision, the concept of explainable models is quite popular. Being able to explain decisions made by a model can be important in the case of BLM as well. A byproduct of explainability could be, for example, a knowledge graph, which is a compact way of summarizing much information, as well as discovering new information [46]. Biological information can be represented in its most general form as knowledge graphs. A model that can be used to curate and represent from new literature entities such as genes, proteins, phenotypes, etc., and their relationships in the form of knowledge graphs can address some issues of the classification approach discussed before. The nodes of the knowledge graph represent the entities, and the edges are the annotations of the directed or undirected relations among the entities. Customized edge annotations, as per the interest of the modeler, can be fed into a model, making the model adaptable to the need of the modeler. Given the positional information on a word representing an edge annotation (e.g., activation, repression, phosphorylation) in a sentence, self-attention-based models can be used to predict the positions of the source and target node entities (e.g., source gene, target gene) for that particular edge annotation. In case a modeler is not interested in modeling interentity relationships in particular and is simply interested in modeling whether there is an association between two entities (gene–phenotype association), such a model can account for this by learning the position of the target entity, given the position of the source entity or vice versa.

A knowledge-graph-based model, as discussed above, could be used in a pipeline with other NLP tasks to develop an end-to-end approach for customized knowledge extraction and knowledge discovery. For example, NER and document triage can be used as preceding tasks in a pipeline. The discovery of relationships among new entities can be achieved through models operating on knowledge graphs generated by the model. For example, Liu et al. proposed a model for the discovery of new relationships among compounds and diseases from knowledge graphs using a reinforcement learning approach on knowledge graphs [46].

## 5. Conclusions

Clearly, attention-based models, both novel architectures and pretrained networks, are being explored widely in the domain of BLM. Complex algorithms have been constructed to handle a wide variety of tasks such as NER, RE, document classification, and triage mining. Some publications have proposed coherent workflows attempting to make the algorithms more practically usable. However, challenges such as diversely annotated datasets, the transfer of knowledge for trained models across datasets, the lack of explainability, complex preprocessing protocols, and the large amount of computational power required to tune pretrained models reveal the scope of further research in this domain with the goal of a more generalistic and practically useful approach.

## Figures and Tables

**Figure 1 biomolecules-11-01591-f001:**
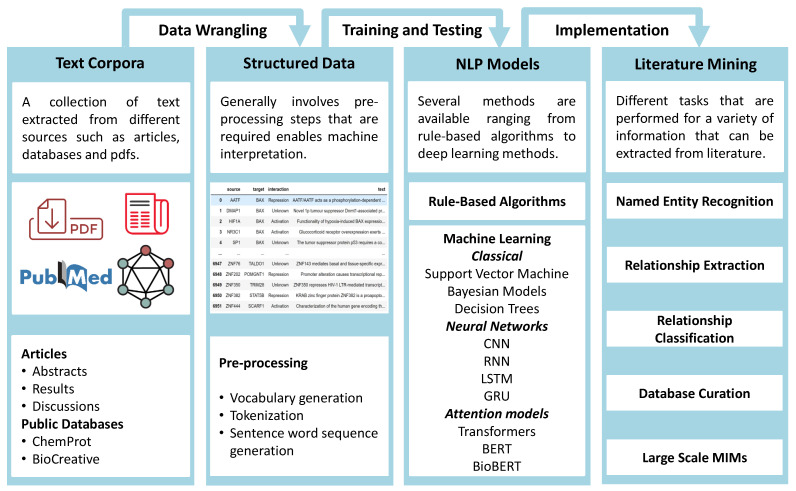
Workflow for Biological Literature Mining (BLM). Starting with a collection of texts from different sources to processing them into structured data for modeling. Choosing from a plethora of NLP models such as BioBERT to perform BLM tasks such as NER and RE to extract information from the text.

**Figure 2 biomolecules-11-01591-f002:**
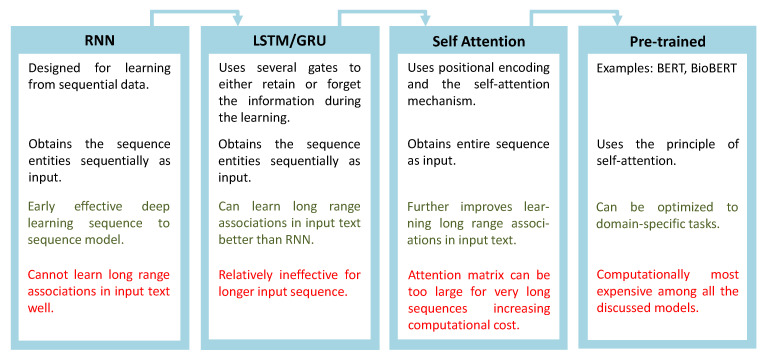
The evolution of sequence-to-sequence models for relationship extraction. Some pros and cons of the models are marked in green and red, respectively.

**Figure 3 biomolecules-11-01591-f003:**
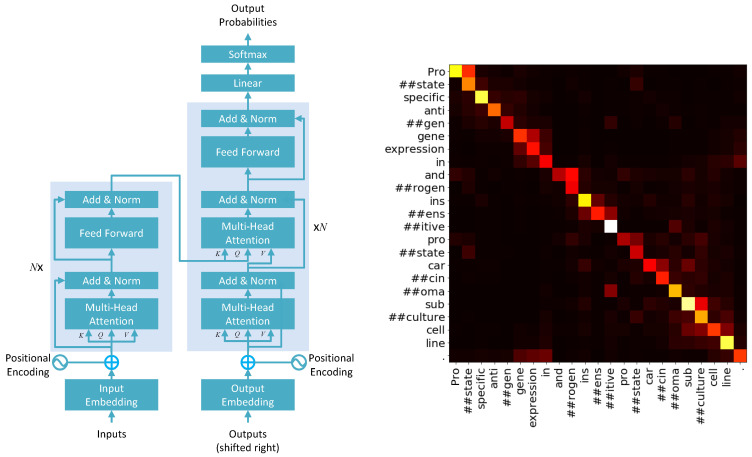
Left: A schematic of the transformer model architecture with attention-based encoder–decoder architecture. The encoder’s output is passed into the decoder to be used as the key and query for the second attention layer. The symbol N× next to the transformer blocks in the encoder and the decoder represents *N* layers of the transformer block. Right: An example heat map of the attention mechanism. The heat map shows pairwise attention weights between pieces of strings in a sentence for a trained model. A hotter hue for a block in the heat map corresponds to higher attention between the string in the row and the string in the column, respective to the block.

**Figure 4 biomolecules-11-01591-f004:**
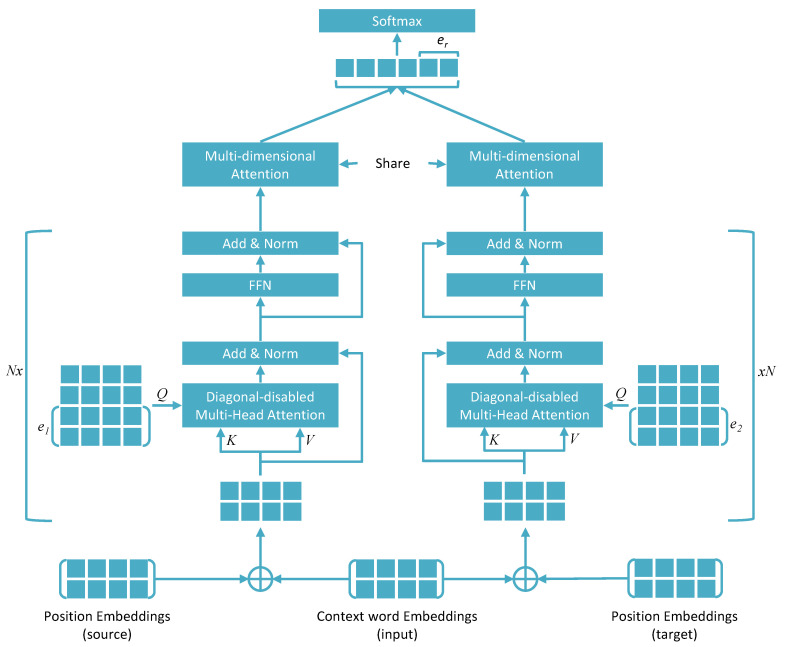
Architecture of the Knowledge-aware Attention Network (KAN). The symbol N× next to the marked blocks represents *N* copies of the respective blocks, where *N* can be defined by the modeler. The architecture has two parallel input channels, taking information on the source and target entity as the input along with the relevant text. External knowledge is integrated into the model in the form of entity-specific representations $e_{1}$ and $e_{2}$ and a representation of their known relationship $e_{r}$.

**Table 1 biomolecules-11-01591-t001:** Key abbreviations for text mining. Rows with text mining problem names are marked in blue; model names are marked in red; biological interaction database names are marked in green.

Abbreviation	Full Name	Description
BLM	Biological Literature Mining	Mining information from biological literature/publications
NLP	Natural Language Processing	Ability of a computer program to understand human language
RE	Relationship Extraction	Extracting related entities and the relationship type from biological texts
NER	Named Entity Recognition	NLP-based approaches to identify context-specific entity names from text
CNN	Convolutional Neural Network	A type of neural network popularly used in computer vision
RNN	Recurrent Neural Network	One of the neural network models designed to handle sequential data
LSTM	Long Short-Term Memory	A successor of RNN useful for handling sequential data
GRUs	Gated Recurrent Units	A successor of RNN useful for handling sequential data
BERT	Bidirectional Encoder Representations from Transformer	A pretrained neural network popularly used for NLP tasks
KAN	Knowledge-aware Attention Network	A self-attention-based network for RE problems
PPI	Protein–Protein Interaction	Interactions among proteins, a popular problem in RE
DDI	Drug–Drug Interaction	Interactions among drugs, a popular problem in RE
ChemProt	Chemical–Protein Interaction	Interactions among chemicals and proteins, a popular problem in RE

**Table 2 biomolecules-11-01591-t002:** Table summarizing several aspects of the compared studies. Several publications investigated different variants of the proposed models. We present the performance of only the best models among them.

Work	Datasets	Model	Tasks Performed	Performance
Elangovan et al. (2020) [39]	Processed version of the IntAct dataset with seven types of interactions	Ensemble of fine-tuned BioBERT models; no external knowledge used	Typed and untyped RE with relationship types such as phosphorylation, acetylation, etc.	Typed PPI: 0.540; untyped PPI: 0.717; metric: F1 score
Giles et al. (2020) [40]	Manually curated from the MEDLINE database	Fine-tuned BioBERT model; used STRING database knowledge during dataset curation	Classification problem with classes coincidental mention, positive, negative, incorrect entity recognition, and unclear	Curated data and BioBERT: 0.889; metric: F1 score
Su et al. (2020) [41]	Processed versions of the BioGRID, DrugBank, and IntAct datasets	Fine-tuned the BERT model integrated with LSTM and additive attention; no external knowledge used	Classification tasks on PPI (binary), DDI (multiclass), and ChemProt (multiclass)	PPI: 0.828; DDI: 0.807; ChemProt: 0.768; metric: F1 score
Su et al. (2021) [42]	Processed versions of the BioGRID, DrugBank, and IntAct datasets	Contrastive learning model; no external knowledge used in dataset curation or as a part of the model	Classification tasks on PPI (binary), DDI (multiclass), and ChemProt (multiclass)	PPI: 0.827; DDI: 0.829; ChemProt: 0.787; metric: F1 score
Wang et al. (2020) [43]	Processed versions of the BioCreative VI PPI dataset	A multitasking architecture based on BERT, BioBERT, BiLSTM, and text CNN; no external knowledge used	Document triage classification, NER (auxiliary tasks), and PPI RE (main task).	NER task: 0.936, PPI RE (exact match evaluation): 0.431; metric: F1 score
Zhou et al. (2019) [44]	Processed versions of the BioCreative VI PPI dataset	KAN; TransE used to integrate prior knowledge from the BioGRID and IntAct datasets on triplets to the model	PPI-RE classification task from BioCreative VI	PPI RE (exact match evaluation): 0.382 PPI RE: (HomoloGene evaluation): 0.404; metric: F1 score

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
