# Peer review of "Self-Attention-Based Models for the Extraction of Molecular Interactions from Biological Texts"

_biomolecules, 2021, doi:10.3390/biom11111591_

Round 1

Reviewer 1 Report

In this review paper, the authors cover self-attention based natural language processing models  applied to molecular interaction extraction, published since 2019. It can serve as an introduction of state-of-the-art automated Biological Literature Mining methods to interested biologists. 

It will be more informative to comment on possible causes of some seemingly low F1-score performance.

Author Response

We thank you for acknowledging our effort to write this article and asking an interesting question. We observed that the F1-Score is naturally less in classification problems with multiple classes. For example, Su et al. has a binary classification task for PPI and thus has a superior F1-Score compared to Elangovan et al. who have considered multiple classes in their classification problem. A multi-class classification problem is inherently more complex. Furthermore, F1-scores are low for models that use exact match evaluation for measuring their performance.

Reviewer 2 Report

Nice overview paper of the latest information extraction techniques with an application to BLM. I miss two things which in my opinion belong to such a survey

  1. Exact description of the way the systematic literature review was performed.
  2. Some way of comparing the different systems. If the field is really as described it seems terribly hard to make progress. Why not take a rather simple dataset with fixed train and test sets, and run all described systems on that task, and produce a leaderboard? Then the contribution of this paper extends that of a mere compilation of summaries. 

Good luck and congratulations with a fine survey!

Author Response

Thank you for the suggestion. We now have the proper rationale on how the study was conducted, added in the manuscript. 

‘Rule-based text mining approaches have been reviewed comprehensively by Zhao et al. Several deep learning-based approaches have been covered by Zhang et al., covering publications until 2019. Self-attention based models entered the stage around 2017; pre-trained networks for the extraction of molecular interactions at an abstract or sentence level, from biomedical texts, like BERT and BioBERT were published after 2018. In this review, we, therefore, focus on self-attention based models, both novel architectures and pre-trained networks, published from 2019 onwards.

For mining recent works on self-attention based models, we searched for published articles and preprints that are publicly available. We searched for well-known databases for publications such as arXiv, bioRxiv, PubMed, Semantic Scholar etc. In addition to these, we also searched for publication databases on recent issues of several related journals and conferences. We only surveyed research articles that involved self-attention mechanism for biological literature mining.’

You suggested a comparative study among the discussed methods. While we would be happy to perform such a study, we must point out some hindrances we face in this regard:

  1. Although most papers have provided code for their models, the implementation turned out to be rather difficult. Most models implement different pre-processing steps that are not clear in the provided codes and the papers. 
  2. These architectures address different sub-tasks of BLM and cover aspects of BLM which are not necessarily similar. For instance, Wang et al. use auxiliary tasks along with Relationship Extraction such as Document Triage and NER for their BioBERT based model, whereas Zhou et al. focus on the Relationship classification task.